# Evaluation of Soil Quality and Analysis of Barriers of Protection Forests along Tarim Desert Highway Based on a Minimum Data Set

**Wenhe Liu [1,2], Lin Li [1,2], Xuemin He [1,2,*] and Guanghui Lv [1,2,*]**

1   College of Ecology and Environment, Xinjiang University, Urumqi 830017, China; liuwenhe@stu.xju.edu.cn (W.L.)
2   Key Laboratory of Oasis Ecology of Education Ministry, Xinjiang University, Urumqi 830017, China
*   Correspondence: hxm@xju.edu.cn (X.H.); ler@xju.edu.cn (G.L.)

**Abstract:** To gain a thorough grasp of the soil quality conditions in the preservation forests along the Tarim Desert Highway, a detailed assessment of the research area's soil quality is necessary, along with the identification of any potential obstacles. This evaluation should identify any potential obstacles and provide a theoretical basis for the sustainable utilization and precise management of protection forest soils. This paper examines the protection forest along the Tarim Desert Highway as the study area. To characterize the features of the soil quality, thirteen indicators of the chemical, physical, and biological composition of the soil were examined. The principal component analysis method was used to construct the minimum dataset (MDS) for soil quality evaluation. The diagnostic model for obstacle factors was combined with the MDS to explore the soil quality characteristics and obstacle factors in the study region. The findings indicated that (1) the constructed indexes of the MDS of protection forest soil in the study area included soil ammonium nitrogen ($NH_4^+$-N); quick-acting phosphorus (AvP); organic carbon (SOC); alkaline phosphatase (AP); and total salt (SS); (2) soil quality based on the MDS and the total dataset (TDS) showed a significant positive correlation ($R^2 = 0.748$; $p < 0.05$), and the research region's soil quality were all in the medium and below level (100% of SQI $\leq$ 0.6), showing the status quo of high soil quality in the two ends of the shelterbelt forest and the tower center area, while the other areas were relatively low; (3) according to the findings of the soil barrier analysis, the two main obstacles influencing the soil quality of the shelterbelt forest at this time are $NH_4^+$-N and SOC, and that the soil quality of the shelterbelt forest could be improved by artificially increasing the inputs of nutrients and precise conservation measures, promoting the overall function of artificial protective forests on desert highways. In addition to providing a solid scientific foundation for the sustainable use and maintenance of shelterbelt forests along the Tarim Desert Highway, this study is also an invaluable resource for researching the soil quality of artificial shelterbelt forests in arid regions.

**Keywords:** MDS; soil quality; Tarim Desert; protection forests

## 1. Introduction

Afforestation and tree planting is one of the most important measures to slow down the movement of sand by wind [1,2]. With planted forests making up almost 38% of the total forest area, China has the biggest area of planted forests worldwide [3,4]. However, the construction of plantation forests can alter the original soil characteristics and quality. Soil plays an extremely important role in improving biological productivity, maintaining the safety of animals and plants, and promoting the healthy development of humans. Soil quality reflects the physical, chemical, and biological functions of the soil [5]. A comprehensive evaluation of soil quality can be achieved by using physical, chemical, and biological indicators [6,7]. The MDS [8] is a collection of smaller indicators that provide richer information, and it is useful for assessing the quality of soil. It has been successfully





applied in the assessment of cropland, grassland, and oasis soil quality [9–13]. Wang Shuqin et al. [14] conducted a study on alfalfa soils in the semiarid zone of the Loess Plateau and established an MDS for soil quality evaluation. The MDS included soil capacity, TN, AvP, UA, fungal Shannon index, and fungal Chao index. The results demonstrated that using this MDS method provided consistent soil quality evaluations compared to using the TDS. Similarly, Gonlu et al. [15] applied an MDS consisting of TN, SS, and catalase to evaluate soil quality in the upper Tarim oasis. However, the application of MDS in evaluating soil quality in shelterbelt forest systems in arid zones has been less frequent [12,13,16]. Therefore, incorporating the MDS in assessing soil quality within shelterbelt forest systems in arid zones can effectively eliminate data redundancy and provide an precise reflection of the current soil quality status.

The soil in the afforestation areas along the Tarim Desert Highway is an important foundation for windbreak and sand fixation plants to grow and reproduce. However, the afforestation in this area is artificially planted and spans the entire Tarim Desert from north to south, making the soil structure and function very fragile with weak resistance to external environmental disturbances. The afforestation along the Tarim Desert Highway ensures smooth traffic on the road, reduces the damage caused by wind and sand to the road, and plays an important role in improving the local microclimate and enhancing soil quality. However, the soil along the highway is seriously desertified and impoverished, lacks surface runoff, heavily relies on groundwater resources, and strongly reflects the water resource distribution pattern dominated by human production and livelihood.

Protective forests in desert areas are established with the aim of improving the quality of desert soil along highways. Therefore, evaluating the soil quality is crucial for ensuring sustainable resource utilization and effective management in arid regions. Both domestic and international scholars have conducted research on individual physical and chemical indicators of soil in protective forests across various regions [17]. However, there remains a lack of comprehensive evaluation regarding shelterbelts, specifically the assessment of soil quality in arid regions [8,18]. Consequently, It is essential to carry out a systematic evaluation of soil in plantation forests within arid zones to enable accurate management of these forests in the future. The purpose of this study is to assess the shelterbelt's soil quality along the Tarim Desert Highway. To enhance the assessment, biological indicators are introduced in addition to the candidate indicators of soil moisture, salinity, and nutrients. The MDS is constructed using a combination of principal component analysis, fuzzy mathematical method, and Norm value. The findings of this evaluation not only serve as a valuable supplement to applied research on ecological engineering in extremely arid areas but also provide insights into the current state of soil quality and the challenges faced by afforestation areas along the desert highway. Moreover, the study offers important insights for the restoration and management of vegetation and soil in the study area, contributing to the stabilization of desert highway operations.

## 2. Materials and Methods

### 2.1. Overview of the Study Area

The protection forest along the Tarim Desert Highway spans between 37°–42° N and 82°–85° E, running through the Taklimakan Desert in a north–south direction. The topography varies along the highway, with the north having lower heights and the south having higher elevations. Overall, the terrain shows minimal undulation, with a relative altitude difference of no more than 80 m. The region experiences a climate characterized by low precipitation, high variability, significant evaporation rates, abundant heat and light, and intense wind and sand activity. Climate conditions exhibit spatial and temporal variations due to an average annual temperature of 12.7 °C, approximately 2900 h of sunshine, and 11 to 50 mm of precipitation per year, primarily falling between May and August. Notably, the yearly potential evapotranspiration can reach 3639 mm. The predominant soil type along the highway is brown–desert soil, with other soil types being less prevalent in the area [19]. The protection forest is predominantly composed of mobile sandy soil, which accounts for

over 85% of the total area. This type of soil has a high fine sand content, low organic matter content [20], salt content of no more than 2‰, a pH value of about 8.5, and soil moisture content of less than 0.5%, posing challenges for plant growth. The selected afforestation species include *Tamarix autromongolica*, *Calligonummongolicum*, *Haloxylon ammodendron*, and other highly resilient trees that excel in fixing wind and sand, as well as demonstrating drought and salt resistance.

### 2.2. Sample Site Layout and Sample Collection

The sampling site was established in the protective forest adjacent to the desert highway, with data collection taking place in July 2023. Ten sample plots, each covering an area of 100 m × 100 m and spaced at 40 km intervals, were designated. Within each plot, ten sample squares were randomly chosen, and soil samples from these squares (0–60 cm depth) were collected. Two portions of soil were obtained: one portion was air-dried and sifted for the analysis of soil physical and chemical properties as well as soil enzyme activities, while the other portion was refrigerated at $-4$ °C for the assessment of soil microbial phosphorus content.

### 2.3. Methodology for Testing and Analyzing Indicators

In order to establish the establishment of a soil MDS, a total of 13 soil candidate indicators were selected in this study. The specific testing method was as follows: soil moisture (SWC) by drying method; bulk density (BD) by ring knife method; pH by potentiometric method; total salt (SS) by conductivity meter (DDSJ-308F, Conductivity Meter, Shangai INESA Scientific Instrument Co., Ltd., Shanghai, China); soil organic carbon (SOC) by potassium dichromate and an external heating method; total nitrogen (TN) by Kjeldahl nitrogen fixation method; total phosphorus (TP) by $HClO_4$-$H_2SO_4$ decoction-molybdenum antimony resist colorimetric method; ammonium nitrogen ($NH_4^+$-N) by leaching-indophenol blue colorimetric method; nitrate nitrogen ($NO_3^-$-N) by phenol disulfonic acid colorimetric method; quick phosphorus (AvP) by $NaHCO_3$ leaching-molybdenum antimony resist colorimetric method; alkaline phosphatase (AP) by colorimetric method using disodium benzene phosphate; urease (UA) by colorimetric method using sodium phenol-sodium hypochlorite; and microbial phosphorus (MBP) by chloroform fumigation method, sodium bicarbonate leaching, and molybdenum antimony resist colorimetric method [21,22].

### 2.4. Soil Quality Assessment Methods

#### 2.4.1. Construction of Minimum Data Set for Soil Quality Evaluation

The 13 soil indicators mentioned above that encompass soil physics, chemistry, and biology were selected for this study, including soil enzyme indicators that are sensitive to disturbance. The 13 candidate indicators were screened by principal component analysis, after which the Norm value was used to obtain the soil MDS indicators. This was performed as follows: the loadings of soil indicators on all principal components with eigenvalues at least equal to 1 were calculated [23]. The soil indicators with loadings not less than 0.5 on the same PC were then assigned to a group, and the soil indicators that may enter multiple groups were assigned to the group with lower correlation [24–26]. The norm value of each indicator within each group was calculated, and the indicators within each group whose norm value falls outside the range of the highest score of 10% were excluded, and then further analyzed to examine the correlation between the selected indicators within each group [15,27,28]. If there was a significant correlation between the selected indicators ($p < 0.05$) in the MDS, the indicator with the largest value of the norm was adopted; if there was no significant correlation between the indicators within the group, all the indicators within the group were retained in the MDS.

The norm value reflects the magnitude of its integrated load and the richness of the soil quality information it contains. The calculation of the norm value fully retains the key

information contained in the indicator while eliminating data duplication. The following formula can be used to determine the norm value [24,29].

$$N_{ik} = \sqrt{\sum_{i=1}^{k} \left( u_{ik}^2 \lambda_k \right)} \tag{1}$$

where $u_{ik}^2$ refers to the loading of the i-th indicator on the k-th principal component; $\lambda_k$ refers to the eigenvalue of the k-th principal component; k refers to the number of principal components with eigenvalue $\geq$ 1; $N_{ik}$ refers to the cumulative factor loading value of indicator i on the k-th principal components.

2.4.2. Soil Quality Evaluation Index

The following is the formula for the SQI [18]:

$$SQI = \sum_{i=1}^{n} W_i \times N_i \tag{2}$$

In the formula, $W_i$ is the indicator weight, $N_i$ is the indicator score, and n is the number of indicators. In particular, the indicator weight represents the percentage of the total correlation coefficients of all assessment indicators divided by the average correlation coefficients between a given indicator and other indicators. Referring to He Yunlin's soil quality classification, the soil quality index was classified into five grades according to the equidistance method [30]. The grades were set up from high to low, with the low grade falling between 0 and 0.2, the lower grade between 0.2 and 0.4, the medium grade between 0.4 and 0.6, the higher grade between 0.6 and 0.8, and the high grade between 0.8 and 1 [31].

The affiliation index was determined by the affiliation function to which the evaluation index belongs, and affiliation functions are generally categorized into ascending and descending types [9,32,33].

The formula for the ascending distribution function is

$$S(x) = \begin{cases} 0.1 & x \leq x_1 \\ 0.9(x - x_1)/(x_2 - x_1) + 0.1 & x_1 < x < x_2 \\ 1.0 & x \geq x_2 \end{cases} \tag{3}$$

The formula for the descending distribution function is

$$S(x) = \begin{cases} 0.1 & x \geq x_2 \\ 0.9(x_2 - x)/(x_2 - x_1) + 0.1 & x_1 < x < x_2 \\ 1.0 & x \leq x_1 \end{cases} \tag{4}$$

The distribution function was chosen based on the results of each soil indicator in the MDS, where $x_1$ and $x_2$ denote the minimum and maximum values of each soil indicator, respectively.

The nonlinear scoring model for soil quality is a normalized conversion of the measured values of soil indicators to a score of 0 to 1, modeled as follows [34]:

$$S_{NL} = \frac{a}{1 + (x/x_0)^b} \tag{5}$$

where $S_{NL}$ is the nonlinear score value, taking the value of 0~1; the indicator's measured value is $x$; the indicator's average value is $x_0$; $a$ is the maximum score ($a$ = 1); and the slope of the equation is $b$. If the indicator is "the bigger the better", then take $-2.5$, if "the smaller the better", then take 2.5 [13,35].

*2.5. Diagnosis of Soil Barrier Factors*

Degree of impairment was calculated applying the following formula, which is based on a diagnostic model of impairment factors:

$$M_{ij} = \frac{P_{ij}W_i}{\sum_{j=1}^{J} P_{ij}W_i} \tag{6}$$

$$M_i = \frac{\sum_{n=1}^{n} M_{ij}}{n} \tag{7}$$

where $M_{ij}$ denotes the barrier degree of the i-th indicator in the j-th sample point, $M_i$ is the average barrier degree of the i-th indicator, and the larger value indicates the stronger barrier of the factor; $P_{ij} = 1 - M_{ij}$ denotes the gap between a soil evaluation indicator and the ideal state of the soil (with the affiliation degree of 1), and the larger value indicates that the indicator is the more unfavorable to soil quality, and $N_{ij}$ denotes the affiliation degree of the i-th indicator in the j-th sample point; and $W_i$ denotes the weight of the i-th indicator; $N_i$ denotes the affiliation of the i-th indicator in the j-th sample point.

*2.6. Data Processing*

The noteworthy variations in soil physicochemical indicators in each sample plot of the protective forest were examined using one-way analysis of variance (ANOVA). For the reason to create the MDS, TDS indicators were chosen using principal component analysis, and Pearson correlation analysis was employed to assess changes in TDS-SQI and MDS-SQI. To examine the association between TDS-MDS and MDS-SQI, regression scores were employed, the coefficient was made using the coefficient of determination ($R^2$), and the results of the analysis were synthesized to select evaluation methods. Statistical analysis of the data was conducted in SPSS 22.0 (IBM SPSS Statistics for Windows, Version 22.0. IBM Corp., Armonk, NY, USA) and Microsoft Excel 2021; principal component analysis, correlation analysis, and normal distribution test were applied in SPSS 22.0, and plotting was conducted in Origin 2022.

**3. Results**

*3.1. Statistical Characteristics of Soil Physical and Chemical Properties and Biological Indicators*

The statistical characteristics of the soil in the desert highway protection forest study area showed (Table 1) that the BD was 1.25 g·cm$^{-3}$, the mean value of SWC was 0.02%, which is an extremely arid area, the mean value of SS was 1.5 g·kg$^{-1}$, and the maximum value was also below the threshold of heavy salinity (5 g·kg$^{-1}$), which is low in salinity, and the mean values of SOC, NH$_4^+$-N, NO$_3^-$-N, TN, TP, and AvP were 2.55 g·cm$^{-3}$, 4.16 mg·kg$^{-1}$, 24.23 mg·kg$^{-1}$, 0.12 mg·kg$^{-1}$ and 1.84 mg·kg$^{-1}$, respectively. The coefficients of variation of the soil index contents in the study area were small, and the overall nutrient contents were low. Among the biological indicators, the contents of UA and AP were 0.08 mg·g$^{-1}$ and 23.13 mg·g$^{-1}$, respectively, and the content of MBP was 0.13 mg·kg$^{-1}$. As shown in Figure 1, it is evident that BD was significantly higher in the central part compared to the southern parts of the study area ($p \leq 0.05$). In contrast, the central region of the study area had lower SWC and UA contents than the northern and southern regions ($p \leq 0.05$); the contents of SS, SOC, NH$_4^+$-N, and NO$_3^-$-N were significantly higher in the northern part than in the central and southern parts ($p \leq 0.05$); TN levels were significantly higher in the north and central than in the south ($p \leq 0.05$), while soil MBP varied significantly among the north, central, and south regions ($p \leq 0.05$), and the content of soil indexes was gradually reduced from the north to the south. AvP showed markedly higher levels in the north compared to the center and south ($p \leq 0.05$), displaying a decreasing then increasing trend; TP and pH, on the other hand, showed an increasing and then decreasing trend, and there were no notable differences among the samples.

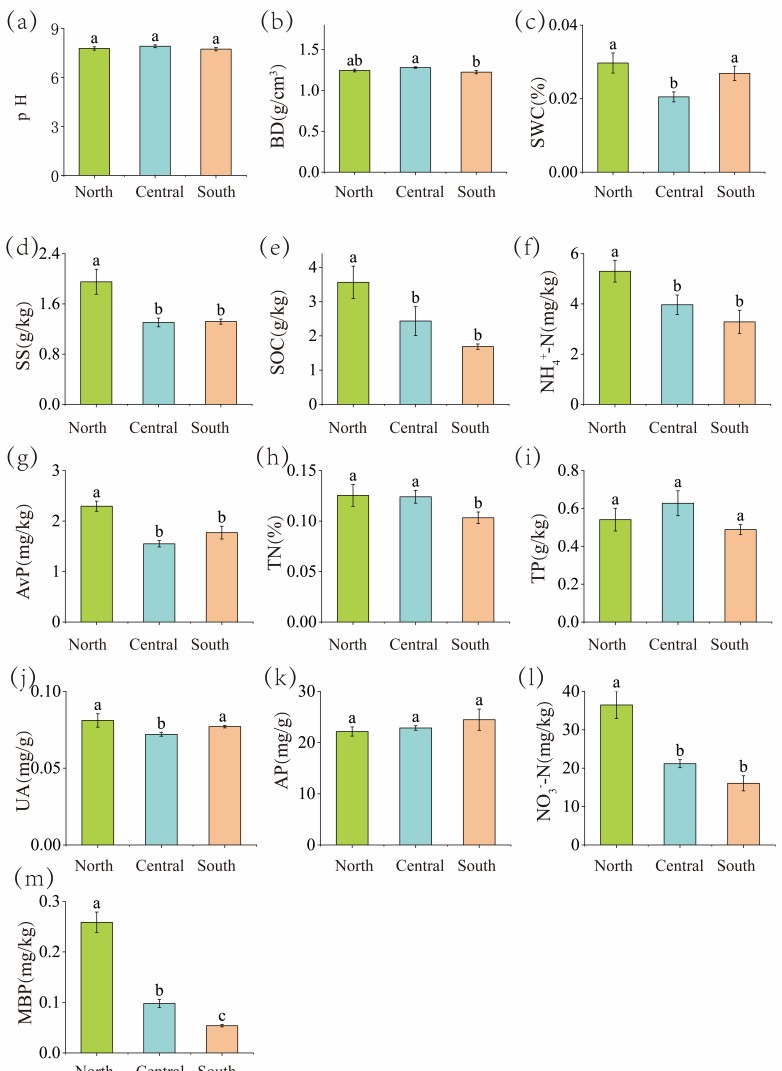

**Figure 1.** Characteristic values of soil indexes in different regions. Note: Different lowercase letters indicate significant differences between different regions of the same indicator ($p < 0.05$); (**a**–**m**) is the physicochemical content of soil in the north, middle and south of the study area.

**Table 1.** Statistical eigenvalues of soil indices.

| Index | Average Value | Maximum Value | Minimum Value | Standard Deviation | Coefficient of Variable |
|---|---|---|---|---|---|
| pH | 7.81 | 8.59 | 6.22 | 0.57 | 0.07 |
| BD | 1.25 | 1.40 | 0.87 | 0.09 | 0.07 |
| SWC | 0.02 | 0.05 | 0.00 | 0.01 | 0.47 |
| SS | 1.50 | 4.78 | 0.35 | 0.72 | 0.48 |
| SOC | 2.55 | 14.63 | 0.89 | 2.31 | 0.90 |
| $NH_4^+$-N | 4.16 | 11.26 | 0.46 | 2.55 | 0.61 |
| $NO_3^-$-N | 24.23 | 63.29 | 5.30 | 15.13 | 0.62 |
| TN | 0.12 | 0.34 | 0.06 | 0.04 | 0.38 |
| TP | 0.56 | 2.09 | 0.36 | 0.33 | 0.59 |
| AvP | 1.84 | 5.15 | 0.54 | 0.62 | 0.34 |
| UA | 0.08 | 0.13 | 0.05 | 0.01 | 0.20 |
| AP | 23.13 | 40.93 | 14.93 | 6.97 | 0.30 |
| MBP | 0.13 | 0.41 | 0.03 | 0.11 | 0.81 |

Note: pH; Bulk density ($g \cdot cm^{-3}$); Soil moisture (%); Soil salt ($g \cdot kg^{-1}$); Soil organic carbon ($g \cdot kg^{-1}$); Ammonium N ($mg \cdot kg^{-1}$); Nitrate N ($mg \cdot kg^{-1}$); Total N (%); Total P ($g \cdot kg^{-1}$); Available P ($mg \cdot kg^{-1}$); Urease ($mg \cdot g^{-1}$); Alkaline phosphatase ($mg \cdot g^{-1}$); MBP ($mg \cdot kg^{-1}$).

### 3.2. Minimum Dataset for Soil Quality Evaluation

The principal component analysis results indicated (Table 2) that the first five principal components with eigenvalues greater than 1 had a cumulative contribution of 74.4% to the variance explained. The contribution of the first principal component to the total variance was 22.12%, with nine indicators, SS, SOC, $NH_4^+$-N, $NO_3^-$-N, TN, TP, AvP, and UA, having high positive loadings, and pH and MBP having high negative loadings; the contribution of the second principal component was 16.28%, with nine indicators, MBP, SWC, $NO_3^-$-N, SS, $NH_4^+$-N, pH, AvP, TN, and SOC, having positive loadings, and UA, BD, and TP having negative loadings; The contribution rate of the third principal component was 12.75%, and AvP, UA, SOC, TP, MBP, and $NH_4^+$-N had high positive loadings. The fourth and fifth primary components had contribution rates of 12.13% and 11.08%, respectively, reflecting the fact that BD, $NH_4^+$-N, MBP, and UA had positive loadings. In addition, the grouping was carried out according to the aforementioned method, and pH, $NH_4^+$-N, TN, and TP were classified into the first group; SWC, SS, $NO_3^-$-N, and MBP were classified into the second group; AvP and UA were classified into the third group; BD and SOC were classified into the fourth group; and AP was classified into the fifth group. According to the Norm values of soil indicators in each group (Table 2) and the results of correlation analysis between indicators (Figure 2), the Norm value of $NH_4^+$-N in the first group was the largest and showed significant correlation with TP, TN, and pH ($p < 0.01$). Therefore, $NH_4^+$-N in the first group of indicators was entered into the MDS; similarly, SS in the second group was entered into the MDS; AvP and SOC were retained in the third and fourth groups, respectively. Only AP was included in the fifth group of indicators; therefore, AP was entered into the MDS. Finally, $NH_4^+$-N (the first group), SS (the second group), AvP (the third group), SOC (the fourth group), and AP (the fifth group) were filtered into the MDS for the soil quality evaluation. Compared with the 13 indicators in the original data, the five selected indicators eliminated data redundancy, and the correlation coefficients of the indicators met the requirements (Figure 2) and accurately represented the soil characteristics of the desert highway shelterbelt forests.

**Table 2.** The principal component factor load and comprehensive loading (Norm) of evaluation indicators.

| Index | Grouping | Matrix of Principal Component Loadings | | | | | Norm Value | MDS |
| --- | --- | --- | --- | --- | --- | --- | --- | --- |
| | | PC1 | PC2 | PC3 | PC4 | PC5 | | |
| pH | 1 | −0.744 | 0.157 | 0.039 | −0.080 | 0.000 | 1.657 | |
| BD | 4 | 0.009 | −0.169 | −0.246 | 0.813 | 0.101 | 1.218 | |
| SWC | 2 | 0.050 | 0.694 | −0.175 | −0.194 | 0.166 | 1.177 | |
| SS | 2 | 0.354 | 0.583 | −0.008 | −0.486 | −0.185 | 1.502 | Y |
| SOC | 4 | 0.457 | 0.108 | 0.403 | 0.649 | −0.092 | 1.571 | Y |
| $NH_4^+$-N | 1 | 0.833 | 0.327 | 0.173 | 0.022 | 0.2.00 | 2.330 | Y |
| $NO_3^-$-N | 2 | 0.118 | 0.648 | −0.073 | 0.109 | −0.534 | 1.367 | |
| TN | 1 | 0.800 | 0.126 | 0.06 | −0.199 | −0.125 | 1.965 | |
| TP | 1 | 0.762 | −0.099 | 0.278 | 0.346 | −0.032 | 2.009 | |
| AvP | 3 | 0.112 | 0.144 | 0.863 | −0.159 | −0.100 | 1.369 | Y |
| UA | 3 | 0.174 | −0.255 | 0.698 | 0.089 | 0.404 | 1.280 | |
| AP | 5 | 0.003 | −0.011 | 0.023 | 0.096 | 0.912 | 1.213 | Y |
| MBP | 2 | −0.119 | 0.769 | 0.236 | 0.006 | −0.114 | 1.404 | |
| Eigenvalues | | 2.876 | 2.117 | 1.658 | 1.577 | 1.440 | 2.876 | |
| Variance contribution rate % | | 22.123 | 16.283 | 12.754 | 12.131 | 11.078 | 22.123 | |
| Cumulative contribution rate % | | 22.123 | 38.406 | 51.161 | 63.292 | 74.369 | 22.123 | |

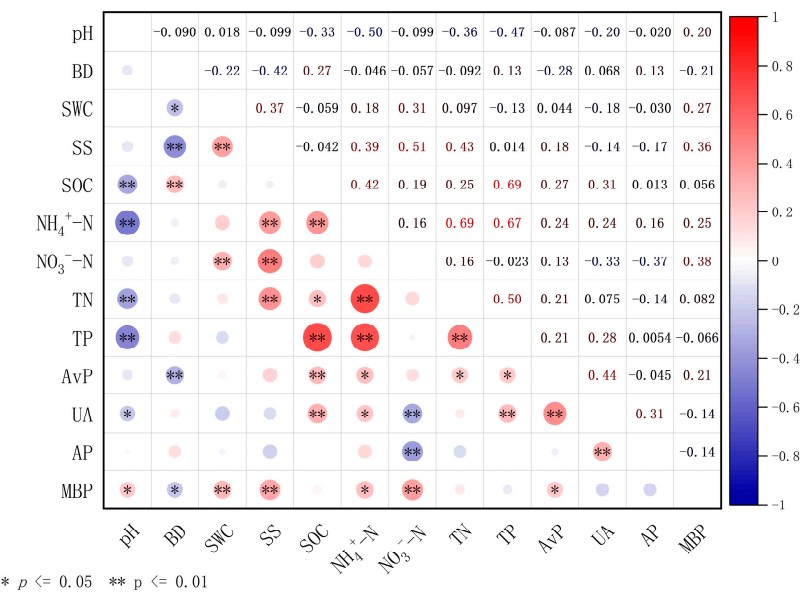

**Figure 2.** Correlation analysis of soil physicochemical indicators.

*3.3. Evaluation of Soil Quality Based on Two Scoring Models*

In this study, the MDS was produced, and principal component analysis was run on it to determine the factor weights and variance of the common factor [9,14] (Table 3). Normalization of MDS metrics was conducted by means of the affiliation function and nonlinear scoring methods. Soil water content, enzyme activities, and soil nutrients in the hinterland of the Tarim Desert are at a low level, and $NH_4^+$-N, AvP, SOC, AP have a high positive effect on soil quality, so it is classified into ascending type of the affiliation function; while soil salinity will play a high negative effect on soil quality, so salinity belongs to the descending type of the affiliation function. Combined with the affiliation function Equations (3) to (4) and Table 3, the final results of soil quality evaluation for different sites in the study area were obtained (Table 4). Meanwhile, the rationality assessment of the MDS and the selection of the evaluation model are important links that affect the accuracy of soil quality evaluation. From Figure 3, it is evident that the assessment of soil quality evaluation using a nonlinear scoring method and affiliation function based on TDS and MDS for soils revealed an extremely relevant positive correlation ($p < 0.01$) between SQI in the same region, i.e., there were similarities in the SQI results between the TDS and MDS based on the two evaluation methods, namely, the nonlinear scoring method and affiliation function, across the northern, central, and southern regions. This indicates the uniformity of TDS and MDS-based soil quality characterization. The regression relationship of soil quality indices was validated using the TDS and MDS (Figure 4), and the regression equations for the nonlinear and the affiliation function scoring methods, respectively, were

$$y = 0.4118x + 0.2859 \quad R^2 = 0.6647,$$

$$y = 0.6486x + 0.1706 \quad R^2 = 0.7483,$$

where x is the MDS-SQI; y is the TDS-SQI. According to the linear regression results, the results obtained by the nonlinear evaluation method and the affiliation function evaluation were significantly positively correlated ($p < 0.01$). The coefficients of determination ($R^2$) of the two methods were 0.66 and 0.74, respectively. The interval of variation of the MDS and the coefficient of variation based on the affiliation function evaluation method (CV = 0.41) were larger than that of the nonlinear evaluation method (CV = 0.30). Therefore, the soil quality of the study area was fitted by the SQI obtained through the affiliation function evaluation, indicating a better fit for the soil quality. The trend of soil quality change was consistent in both the TDS and the MDS, showing sensitivity to changes in SQI and enabling

the evaluation of the comprehensive quality of the soil in the highway protection forests of the Tarim Desert. The integrated evaluation results of soil quality of the MDS based on the affiliation function (Table 4) were ranked as follows: Sample 4 (0.547) > Sample 1 (0.522) > Sample 8 (0.516) > Sample 9 (0.511) > Sample 10 (0.452) > Sample 3 (0.438) > Sample 5 (0.419) > Sample 2 (0.365) > Sample 6 (0.362) > Sample 7 (0.354) ($p > 0.05$). The SQI of the study area from north to south initially decreased, then increased to the highest point at Sample 4, followed by a decrease to the lowest point at Sample 7, and, subsequently, a sharp increase followed by a slow decrease. Overall, the presentation was characterized by high values at both ends and in the center, and lower values in the remaining areas.

**Table 3.** Weights of soil quality evaluation indicators.

| Index | Total Data Set | | Minimum Data Set | |
|---|---|---|---|---|
| | Common Factor Variance | Weight | Common Factor Variance | Weight |
| pH | 0.586 | 0.067 | | |
| BD | 0.760 | 0.037 | | |
| SWC | 0.580 | 0.051 | | |
| SS | 0.736 | 0.037 | 0.685 | 0.080 |
| SOC | 0.813 | 0.133 | 0.573 | 0.285 |
| $NH_4^+$-N | 0.871 | 0.146 | 0.703 | 0.314 |
| $NO_3^-$-N | 0.737 | 0.034 | | |
| TN | 0.714 | 0.075 | | |
| TP | 0.789 | 0.117 | | |
| AvP | 0.813 | 0.073 | 0.381 | 0.157 |
| UA | 0.754 | 0.087 | | |
| AP | 0.841 | 0.076 | 0.594 | 0.163 |
| MBP | 0.674 | 0.067 | | |

**Table 4.** Soil quality in different regions.

| | SQI | | SQI | | SQI |
|---|---|---|---|---|---|
| Plots 1 | $0.522 \pm 0.14$ a | Plots 4 | $0.547 \pm 0.14$ a | Plots 8 | $0.516 \pm 0.16$ a |
| Plots 2 | $0.365 \pm 0.21$ a | Plots 5 | $0.419 \pm 0.19$ a | Plots 9 | $0.511 \pm 0.16$ a |
| Plots 3 | $0.438 \pm 0.13$ a | Plots 6 | $0.362 \pm 0.22$ a | Plots 1 | $0.452 \pm 0.21$ a |
| | | Plots 7 | $0.354 \pm 0.23$ a | | |
| North (0.442) | | Central (0.421) | | South (0.493) | |

Note: Soil quality evaluation of the northern, central, and southern regions shown in parentheses; a indicates no significant change; a indicates no significant difference between plots.

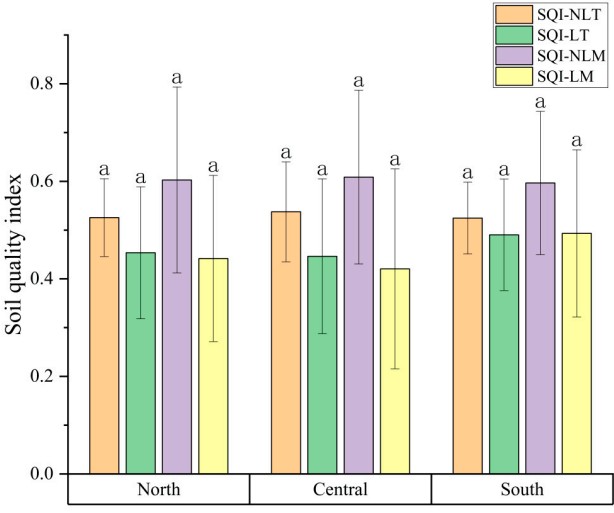

**Figure 3.** Soil quality index based on TDS and MDS. Note: a indicates no significant change.

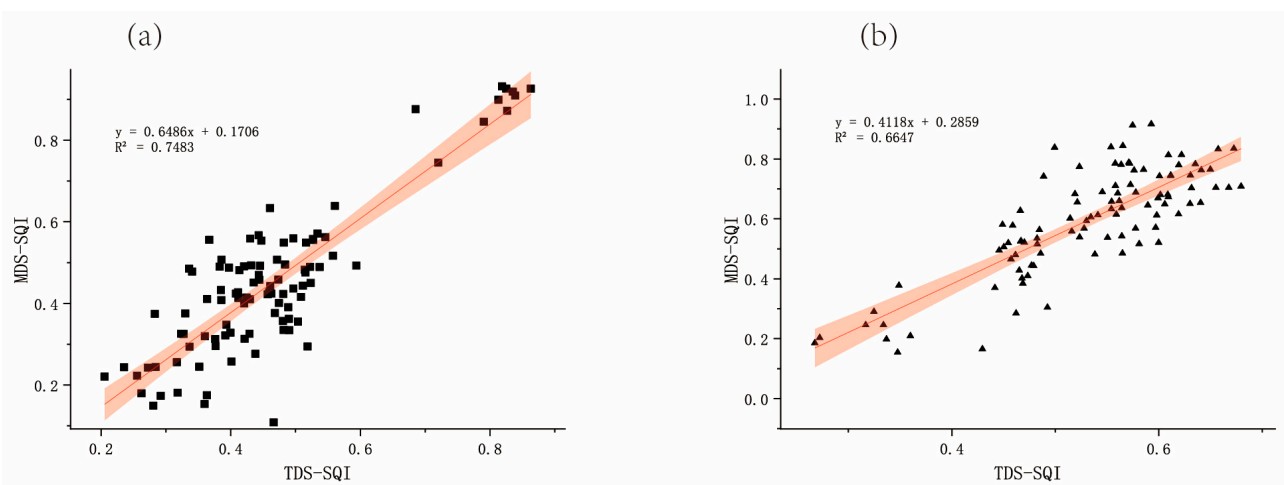

**Figure 4.** Correlation of soil quality index based on TDS-SQI and MDS-SQI. (**a**) Evaluation of membership degree function; (**b**) nonlinear evaluation; The solid red line is the linear fitting line; The black rectangle and triangle are MDS-SQI values.

### 3.4. Analysis of Soil Quality Barriers

The diagnostic model for barrier factor analysis (Figure 5) indicated that the overall soil barrier size in the study area followed this order: $NH_4^+$-N (0.346) > SOC (0.239) > AP (0.179) > AvP (0.161) > SS (0.075). Specifically, for each sample plot, $NH_4^+$-N and SOC were identified as the top two barrier factors. Among these, SOC was the primary obstacle factor in sample areas 2, 6, and 7, whereas $NH_4^+$-N took precedence in the remaining areas. Additionally, the soil salinity obstacle degree was the smallest across all areas of the study site. Considering the variation in the mean size of the barrier degree for each index, it is evident that $NH_4^+$-N and SOC are the most influential factors affecting the soil quality of the desert highway protection forest.

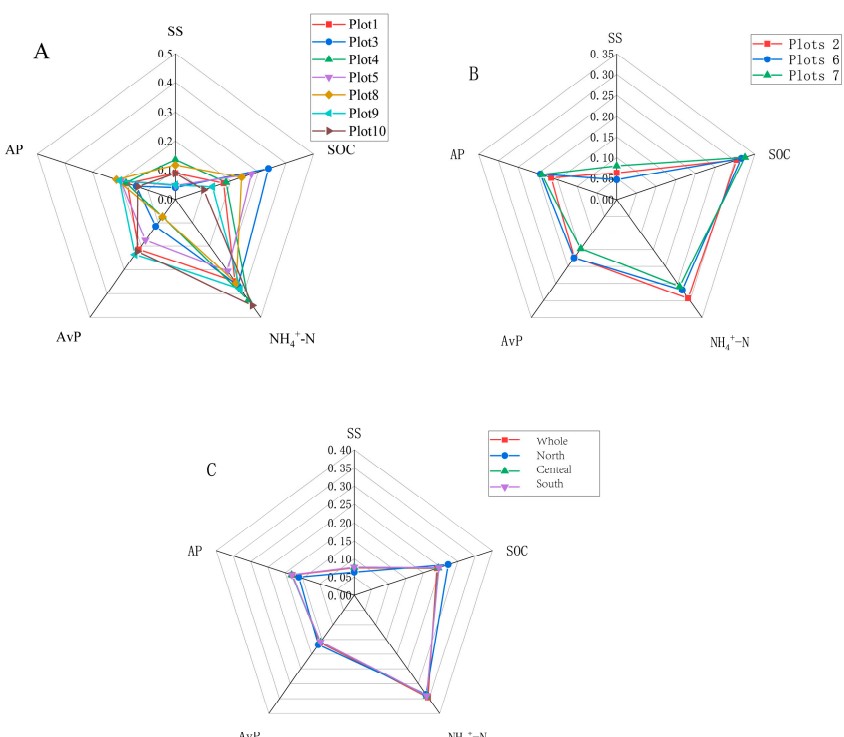

**Figure 5.** Barriers to soil indicators. Note: (**A**,**B**) represent the barrier degrees of plots 1 to 10; (**C**) represents the radar chart of barrier degrees in different regions.

## 4. Discussion

Due to the unique soil conditions in arid regions, the quality of desert soil is generally very poor. Furthermore, soil quality fluctuations can ultimately be attributed to changes in external material energy inputs [3,36–38]. The soil quality environment of arid zone soils after plant colonization is highly enhanced compared to the original bare soil [39–42]. A preliminary analysis of the physical, chemical, and biological indicators of soil in the protection forest has helped to clarify the current state of soil quality along the Tarim Desert Highway. The results show that the chemical and physical characteristics of the soil in different geographic locations of the desert highway do not have complete consistency, which is in line with the conclusions of Jin Zhengzhong's [43–45] research on the soil of the desert highway. According to the National Soil Nutrient Content Classification Standard, the soil nutrient and enzyme contents in this study were at a low or very low level. Relevant scholars [46–49] also pointed out that in the hinterland of the extremely arid Taklamakan Desert, the establishment of artificial vegetation using brackish water irrigation promoted the development of wind–sand soils, which led to significant changes in the soil interior, improved soil physical and chemical properties, and increased fertility. Soil properties affect soil aeration, water permeability, nutrient transport, and other characteristics, which is an important reflection of soil fertility. Guo Ke et al. [50] showed that the soil moisture content after planting was not significantly different from that of quicksand, which is also consistent with the study in this paper. They showed that the physical properties of soil were greatly improved after the planting of protection forests, which favored the water retention of soil after irrigation of vegetation and promoted the formation of soil structure. Cao Chengyou et al. found that the restoration and construction of artificial vegetation under artificial brackish drip irrigation conditions in the flowing dunes of the Horqin Sandland benefited the improvement of soil nutrient levels and biological activity [51–53]. Organic matter is one of the most important indicators of soil fertility [54]. It is the difference between soil organic matter inputs and their utilization by plants and microorganisms. It is the most important carbon source of the soil, which has a direct influence on the nutrient supply and antidisturbance of the soil. Nitrogen and phosphorus are frequently limiting factors in soil fertility. Nitrogen's inertness and phosphorus's insolubility and immobility contribute to this limitation. Enzyme activities are responsive to soil conditions. Based on the results of the barrier analysis, SOC, $NH_4^+$-N, and AvP are key factors that must be addressed to ensure stable operation of the shelterbelt, and emphasizing and addressing these key barriers is critical to soil quality of the study area.

Soil is formed by the integrated effects of topography, climate, and human activities, among other factors, and changes over time with ecological succession. Soil quality is often evaluated using statistical methods such as cluster analysis, principal component analysis (PCA), and correlation analysis (CA) to construct an MDS and screen out redundant indicators. Among these methods, PCA is undoubtedly the most widely used for constructing an MDS. In this study, five soil indicators, namely, SS, SOC, $NH_4^+$-N, AvP, and AP, were selected from 13 candidate indicators by principal component analysis and Norm value to construct an MDS for comprehensive evaluation of spatial changes in soil quality of protection forests along Tarim Desert Highway. The soil indicators selected by scholars for the MDS in evaluating soil quality across various research subjects can vary significantly [33]. However, these results typically encompass a range of physical, chemical, and biological indicators. SOC/SOM and SS stand out as the most commonly utilized indicators in soil quality assessment [55]. Following these, AvP and other indicators are often employed, while TN, TP, and water content are also frequently considered in soil quality evaluation. Tian Ying et al. [56] selected the indicators of SOM, TN, and TP into the MDS when assessing the condition of the soil in several planting woods situated in the sandy Yinchuan Plain in Ningxiad. Li Peng et al. [57] screened the indicators of TN, sand, TP, pH, and effective water content to construct the MDS for assessing Beijing's forest land's soil quality; and Qiao Yunfa et al. [58] screened the MDS with BD, pH, and AvP as the evaluation indexes for evaluating the quality of tillage soil in the northeast windy sandy

soil. From the previous studies, most of the soil indexes used in the research were biased towards physical and chemical properties of the soil, and most of the biological indexes were microorganisms, which were used to evaluate the quality of the soil [12,33]. However, in recent years, soil enzyme activities have been emphasized by researchers because they can more sensitively reflect changes in the soil environment, and have been widely accepted to be included in various soil quality evaluation systems. Cao Xiaoyu et al. constructed an MDS with BD, TN, TP, AK, pH, and CAT as indicators for soil quality evaluation of Chinese cedar plantation forests, and the results of MDS and TDS were consistent. Gong Lu et al. [15] established an MDS with TN, SS, and catalase as factors to evaluate soil quality in the upper Tarim oasis. Zhu et al. [59] constructed an MDS containing sucrase, CP, TP, and AN to evaluate the soil quality of Eucalyptus plantation forests. Ammonium nitrogen is an important source of nitrogen for plant uptake; In dry zones, salinity is a significant element that restricts plant growth and development; SOC is an important nutrient element for plant growth; AvP is the most readily available phosphorus for plant uptake and utilization; and AP has an important effect on phosphorus nutrition and reflects the intensity of microbial activity. In this study, the five soil indicators in the minimum dataset screened for evaluating the soil quality of protection forests along the Tarim Desert Highway are very appropriate. Prior research has extensively employed these five indicators to assess soil quality at various scales in shelterbelts [60].

In the northern part of the research area, soil quality ranges from 0.438 to 0.522, averaging 0.445 overall. Sample site No. 1 has the highest soil quality at 0.522 among northern sites, while others are lower. This is mainly because the northernmost area is near the edge of the Tarim Desert, by the Tarim River, and is where shelterbelt forests start. These areas undergo artificial and scientific management, including continuous fertilizer and drip irrigation to enhance soil fertility and water content. The soil environment has been effectively improved and the soil quality is relatively high because of the scientific artificial management, continuous fertilization and drip irrigation to increase soil fertility and improve soil moisture content and distribution, as well as the reduction in soil salinity and the improvement of soil nutrients and soil enzyme activities. With an SQI of 0.420, the soil quality in the research area's center was the lowest. Sample Plot No. 4 had the highest soil quality in this area (SQI = 0.522), while Sample Plot No. 7 had the lowest (SQI = 0.354). This difference in soil quality can be attributed to human production. Sample Plot No. 4 is located near the town of Tazhong and the Tazhong oil field management area, where manmade management is intense, with a high degree of vegetation cover, diversity, and intensity [61]. In contrast, the protective forest belt deep in the desert hinterland has deteriorating soil quality, with the lowest rating in Sample Site 7 (SQI = 0.254). Although the protective forest is irrigated and receives consistent nutrient inputs, damage from sand and wind is caused by the harsh environment in the Tarim Desert's hinterland [62,63], resulting in soil structure destruction, topsoil loss, plant death, cover loss, soil resanding and salinization, soil water and nutrient loss, decreased enzyme activity, and rapid soil quality degradation [47,48,54,64]. In the southern part of the research region, the soil quality of the protection forest is notably high, averaging 0.493. Specifically, Sample Plot No. 8 and Sample Plot No. 9 exhibit relatively high soil quality levels. This is attributed to reduced wind speeds in the southern area, resulting in decreased soil movement and a more stable soil environment. Additionally, being adjacent to the Niya River Basin and the Lower Yathungus River–Lower Andil River Basin contributes to increased groundwater replenishment, thereby enhancing soil water content and overall quality [65–68]. Under the management mode of mainly manual management, the soil quality of the study area is medium and below (100% of SQI ≤ 0.6). The management mode of protection forests in low-value areas can differ from that of high-value areas. In low-value areas with lower soil quality, inputs of soil moisture and nutrients will be increased based on actual conditions. Additionally, areas with dead vegetation and low cover will be replanted to improve their condition. For areas with low soil quality, the input of soil water and nutrients can be increased according to their actual conditions, replanting plants in areas with dead

vegetation and low coverage, and effectively controlling the speed of wind and sand movement in the protective forest belt, so as to reduce the destruction of the surface soil structure and the loss of nutrients.

### 5. Conclusions

In this study, it was found that the protective forest $NH_4^+$-N, AvP, SOC, AP, and SS could better characterize the soil quality in the study area, and could be used as the MDS to evaluate the soil quality of the protective forest along the Tarim Desert Highway. When calculating the SQI in a study area, adopting an evaluation method based on membership functions can more accurately capture the contribution of each indicator to soil quality. This approach can provide superior results compared to the nonlinear function method, thereby offering significant advantages. The soil quality was high at the two ends and in the central of the Tarim Desert, and relatively low in the other areas; the soil quality in the study area was at the medium level or below. Based on the analysis of soil barriers in the protection forests at the present stage, the findings indicate that soil $NH_4^+$-N and SOC are currently the two primary factors influencing the quality of protective forest soils. In order to make the Tarim Desert Highway shelterbelt forest more effective and sustainable, the shelterbelt soil should be managed more precisely and differently, and targeted resource inputs and management should be adopted for the areas with low soil quality.

**Supplementary Materials:** The following supporting information can be downloaded at: https://www.mdpi.com/article/10.3390/land13040498/s1.

**Author Contributions:** Conceptualization, W.L.; methodology, W.L.; software, W.L.; formal analysis, W.L. and L.L.; investigation, W.L.; writing—original draft preparation, W.L.; writing—review and editing, W.L., X.H. and G.L.; visualization, W.L.; supervision, X.H. and G.L.; funding acquisition, G.L. All authors have read and agreed to the published version of the manuscript.

**Funding:** This work was funded by the determination of carbon sink in sand-fixing forest belt in Luntai–Minfeng section of desert highway (202205140011).

**Institutional Review Board Statement:** Not applicable.

**Informed Consent Statement:** Not applicable.

**Data Availability Statement:** Data are contained within the article and Supplementary Material.

**Conflicts of Interest:** The authors declare no conflicts of interest.

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
