# Peer review of "Evaluation of Soil Quality and Analysis of Barriers of Protection Forests along Tarim Desert Highway Based on a Minimum Data Set"

_land, doi:10.3390/land13040498_

Round 1
Reviewer 1 Report
Comments and Suggestions for Authors
The manuscript is well-written and provides sufficient data to support findings. Methodologies are also good enough and can be accepted in the Journal after some revision.
All the corrections are marked in the attached modified manuscript.

Reviewer 2 Report
Comments and Suggestions for Authors
Under the hipotesis that the construction of plantation forests can alter the original soil characteristics and quality, and that evaluating the soil quality is crucial for ensuring sustainable resource utilization and effective management in arid regions, the authors conducted a study to assess the shelterbelts'soil quality along the Tarim Desert Highway. In my opinion the manuscript in its present form should be improved; some paragraphs could be improved as suggested below.
To characterize the features of the soil quality, they use a considerable number of indicators (chemical, physical, and biological composition of the soil); Specifically thirteen were examined. Following current criteria, the main component analysis method was used to construct the minimum data set (MDS) for soil quality evaluation. All this is evaluated positively.
Line 28. What does this isolated phrase mean?: Promote the overall function of desert highway artificial protection forests.
The introduction was too long without outstanding the highlights and scientific problem. Objectives must be clearly stated
Line 98. The predominant soil type along the highway is wind-sand soil, with other soil types being less prevalent in the area. In my opinion, in a work in which soil quality is evaluated, it is very poor not to indicate what types of soil there are. I suggest to add the types of soils following at least one of the most used taxonomies WRB FAO or Soil taxonomy.
Line 103. PetroChina Natural Gas Co., Ltd. is responsible for the construction and management of these protective forests, utilizing drip irrigation. This does not add any additional value to the work. I suggest delete.
Line 116. soil microbial phosphorus content. Please, justify why this parameter is chosen and not another.
The procedure to determine the quality of the soil (2.4.) is correct.
Figure 1 is of poor quality. Please improve it.
Table 4 looks bad.
In the discusion section the lengthy sentences may be split in to smaller sentence without change of its meaning. For example from line 334 to 368.
Line 451. In this study, the soil of protection forest along Tarim Desert Highway was selected 452 as the research target, and the soil quality of protection forest along Tarim Desert 453 Highway was evaluated by constructing the MDS including enzyme activity, which 454 reduced the data redundancy to a larger extent and identified its obstacle factors. Please rewrite this sentence.
My major concern on this paper was that some parts of the manuscript, especially the discussion section, are simple repetitions of other authors; It is also too cumbersome, too long, and above some speculative.
I suggest authors integrate a workflow to assist readers with easy navigation or easily visualize the entire work.
Kindly integrate the relevance or importance of study findings to the international scientific community, as well as local authority or industrial players.
I wish those changes will contribute to improve your paper.
Reviewer 3 Report
Comments and Suggestions for Authors
Please see attached annotated pdf document with comments for the authors.
This is a very interesting and relevant topic, but the authors could have tried to pull this through to their Discussion and Conclusion. More should be made of the fact that semi-arid areas lack MDSs for soil quality assessment, and that this study provides unique insight into this.

First art of the manuscript reads well, however, some attention is needed in the Discussion and Conclusion sections on very long and thus incoherent sentences.
Round 2
Reviewer 2 Report
Comments and Suggestions for Authors
I went through the revised version of the manuscript and found that it had considerably improved from the original manuscript. The language and structure of the manuscript have improved and are more understandable. Therefore, the article is suitable for publication in its present form.